# Expanding the repertoire of glucocorticoid receptor target genes by engineering genomic response elements

Verena Thormann, Laura V Glaser , Maika C Rothkegel, Marina Borschiwer, Melissa Bothe, Alisa Fuchs , Sebastiaan H Meijsing

The glucocorticoid receptor (GR), a hormone-activated transcription factor, binds to a myriad of genomic binding sites yet seems to regulate a much smaller number of genes. Genome-wide analysis of GR binding and gene regulation has shown that the likelihood of GR-dependent regulation increases with decreased distance of its binding to the transcriptional start site of a gene. To test if we can adopt this knowledge to expand the repertoire of GR target genes, we used CRISPR/Cas-mediated homology-directed repair to add a single GR-binding site directly upstream of the transcriptional start site of each of four genes. To our surprise, we found that the addition of a single GR-binding site can be enough to convert a gene into a GR target. The gain of GR-dependent regulation was observed for two of four genes analyzed and coincided with acquired GR binding at the introduced binding site. However, the gene-specific gain of GR-dependent regulation could not be explained by obvious differences in chromatin accessibility between converted genes and their non-converted counterparts. Furthermore, by introducing GR-binding sequences with different nucleotide compositions, we show that activation can be facilitated by distinct sequences without obvious differences in activity between the GR-binding sequence variants we tested. The approach to use genome engineering to build genomic response elements facilitates the generation of cell lines with tailored repertoires of GR-responsive genes and a framework to test and refine our understanding of the *cis*-regulatory logic of gene regulation by testing if engineered response elements behave as predicted.

## Introduction

*Cis*-regulatory elements embedded in the genome encode the information to relay environmental and developmental cues into specific patterns of gene expression. The information is decoded by transcription factors (TFs), which bind to *cis*-regulatory elements and set in motion a cascade of events to change the expression level of genes. Typically, *cis*-regulatory elements harbor clusters of binding sites for several different TFs, and the combinatorial nature of the response element allows different outcomes depending on which combination of TFs is active in a given cell type or under given environmental conditions (reviewed in reference 1). Divergence in gene regulation may also play a role in adaptation and speciation (2) and can be driven by the loss or gain of TF-binding sites that can occur rapidly over evolutionary time (3).

Ligand-activated TFs, such as the glucocorticoid receptor (GR), represent an attractive model TF to study the link between TF binding and gene regulation. An appealing feature of GR to study gene regulation is that its activity can be turned on or off by the addition or removal of its ligand (e.g., dexamethasone, a synthetic glucocorticoid ligand). This on/off switch facilitates the relatively straightforward identification of target genes by comparing gene expression levels between untreated cells and cells treated with hormone. Studies with GR have shown that the genes regulated by GR differ considerably between cell types (4, 5). Accordingly, the genomic loci bound by GR show little overlap between different cell types (4, 6, 7). Furthermore, cross-talk with NFκB signaling can alter the repertoire of genomic loci bound and of the genes regulated by GR (8, 9). The sequence composition of *cis*-regulatory elements also plays a role in fine-tuning the expression level of individual genes. For instance, GR binds as a dimer to typically imperfect half sites separated by a 3-bp spacer, and the exact sequence of the half site, the spacer, and of the nucleotides flanking the GR-binding sequence (GBS) can modulate GR's activity towards target genes (10, 11).

GR can bind to tens of thousands of genomic binding sites, yet seems to regulate a smaller number of genes (4, 7, 12, 13). Part of the discrepancy between GR binding and gene regulation might be technical, for example, because of false positives in ChIP-seq peak calling and false negatives when the criteria for calling genes regulated are too stringent. Furthermore, gene regulation is typically only sampled at a few time points and relies on the analysis of steady-state RNA, which can yield false positives and false negatives, for example, when changes in transcription rates are masked by changes in RNA stability. The discrepancy between GR binding and gene regulation might also be due to GR's inability to activate

Max Planck Institute for Molecular Genetics, Berlin, Germany

Correspondence: meijsing@molgen.mpg.de

gene expression for a subset of occupied sites (14) and could reflect the inability of distal GR-binding sites to contribute to gene regulation because they lack the physical proximity to the promoter of a gene. Accordingly, the link between GR binding and gene regulation is especially weak for ChIP-seq peaks located at large distances from the promoter of genes except when the three dimensional organization of the genome brings these distal GR ChIP-seq peak proximal to the promoter of a gene (13). Nonetheless, even when taking three-dimensional genome organization into account, GR binding is a poor predictor of GR-dependent gene regulation with only a subset of binding events (<25%) resulting the in the regulation of associated genes (13). Recent advances in genome editing now offer opportunities to assay the contribution of endogenous TF-bound regions to gene regulation. For example, by perturbing TF-bound regions in their endogenous genomic context, their contribution to gene regulation can be assessed (15). Similarly, catalytically inactive Cas9 fused to repressor domains, for example, Krüppel-associated box, can be targeted to candidate *cis*-regulatory elements to assess their regulatory function in the genomic context (15, 16). Finally, a fine-grained dissection of the interplay of TF-binding sites within *cis*-regulatory elements can uncover operating principles of active regulatory elements, for example, that a cluster of GR-binding sites is required for the activity of an individual enhancer located near the GR target gene *GILZ* (13). A complementary, largely unexplored, way to study TF-binding sites is to use genome editing combined with homology-directed repair (HDR) to build functional response elements. By building synthetic *cis*-regulatory elements, the minimal sequence requirements for a functional response element and their ability to recapitulate existing expression patterns can be researched.

As described above, genome-wide approaches and perturbation of endogenous response elements can be used to identify operating principles of functional GR-binding sites. One approach to test the validity of these findings is to determine if we can "engineer" *cis*-regulatory elements based on these principles in the genomic context. One of the principles we identified is that the likelihood of GR-dependent regulation increases with decreased distance of its binding site to the transcriptional start site (TSS) of a gene (13). To test if adding a single GR-binding site is sufficient to convert genes into GR targets, we used HDR-mediated genome editing to generate cell lines with a single GBS immediately upstream of their TSS. In addition, we compared GBS variants to test if the sequence identity of the binding site influences GR-dependent gene regulation. Together, our studies reveal that addition of a single GBS can be sufficient to convert genes into GR targets without obvious differences in the level of activation between GBS variants.

# Results

## Addition of a single promoter-proximal GBS can render a gene GR-responsive

To study what is required to convert endogenous genes into GR targets, we first set out to add a single GBS near the TSS of four candidate genes using CRISPR/Cas9 and HDR templates (Fig 1A). To increase our chances of observing GR-dependent regulation, we picked a GBS variant (CGT, a synthetic sequence matching the consensus motif), which showed the highest GR-dependent activation in previous studies (10, 11). The candidate genes were selected based on the following criteria. First, we chose genes that are not regulated by GR and display low basal levels of expression (Fig 1). This was motivated by studies showing that ectopic activation using CRISPRa (CRISPR activation) works best for genes with low expression levels (17, 18). Second, to increase our chances of obtaining correctly edited clones with a TSS-proximal GBS, we only considered genes with a possible guide RNA located ≤50-bp upstream of its TSS, given that HDR efficiency decreases with increased distance between the cut site and the mutation (19). We chose to place the GBS close to the TSS because proximal GR-bound regions are more likely to influence the expression of nearby genes than their distal counterparts (13). Third, we selected guide RNAs with high computationally predicted specificity and low off-target scores (20) and prioritized genes for which a low number of nucleotide exchanges were needed to introduce a GBS. Finally, to increase HDR efficiency, we chose candidates for which introduction of the GBS resulted in PAM-blocking or guide-blocking mutations (19).

Using this approach, we selected single-cell–derived clonal lines for which one allele harbored the engineered GBS ≤ 50-bp upstream of the TSS of four genes (*GYPC*, *IL1B*, *IL1R2*, and *VSIG1*, Figs 1F and S1) in U2OS cells stably expressing GR (U2OS-GR, (21)). As expected, basal expression levels in the absence of hormone were unaffected by the introduced GBS for each of the genes analyzed (Fig 2). Next, we tested if the addition of a single GBS was sufficient to convert the nearby gene into a GR target and observed a robust increase in transcript levels for both the *IL1B* and IL1R2 genes upon treatment with the synthetic glucocorticoid dexamethasone (Fig 2B and C). The activation of the *IL1B* and IL1R2 genes was observed for each of multiple independent clonal lines with an added GBS that we tested (Fig 2B and C). Furthermore, no activation was observed for parental U2OS-GR cells or for unedited clonal controls (Fig 2B and C) showing that the observed activation is a consequence of the GBS addition. For the other two genes we edited (*GYPC* and *VSIG1*), the GBS addition did not convert the gene into a GR target (Fig 2A and D). Together, these experiments show that the addition of a single GBS near the TSS of a gene can be sufficient to convert it into a GR target.

## Gene-specific GR binding partially explains the gene-specific acquirement of GR-dependent regulation upon GBS addition

To test if locus-specific GR binding could explain the gene-specific acquired activation, we analyzed GR binding at the added GBSs by chromatin immunoprecipitation (ChIP). Consistent with the acquired GR-dependent activation, we found that GR was recruited to the *IL1R2* and *IL1B* loci upon hormone treatment for clonal lines with an added GBS, whereas no binding was observed for unedited control cells (Fig 2F and G). For the *GYPC* gene, the ChIP-seq data for the parental U2OS-GR shows a small peak immediately upstream of its TSS (Fig 1B). Accordingly, we find a modest hormone-dependent recruitment of GR to the TSS of *GYPC* for unedited cells, which was slightly higher for the clonal line with an added GBS (Fig 2E). For the

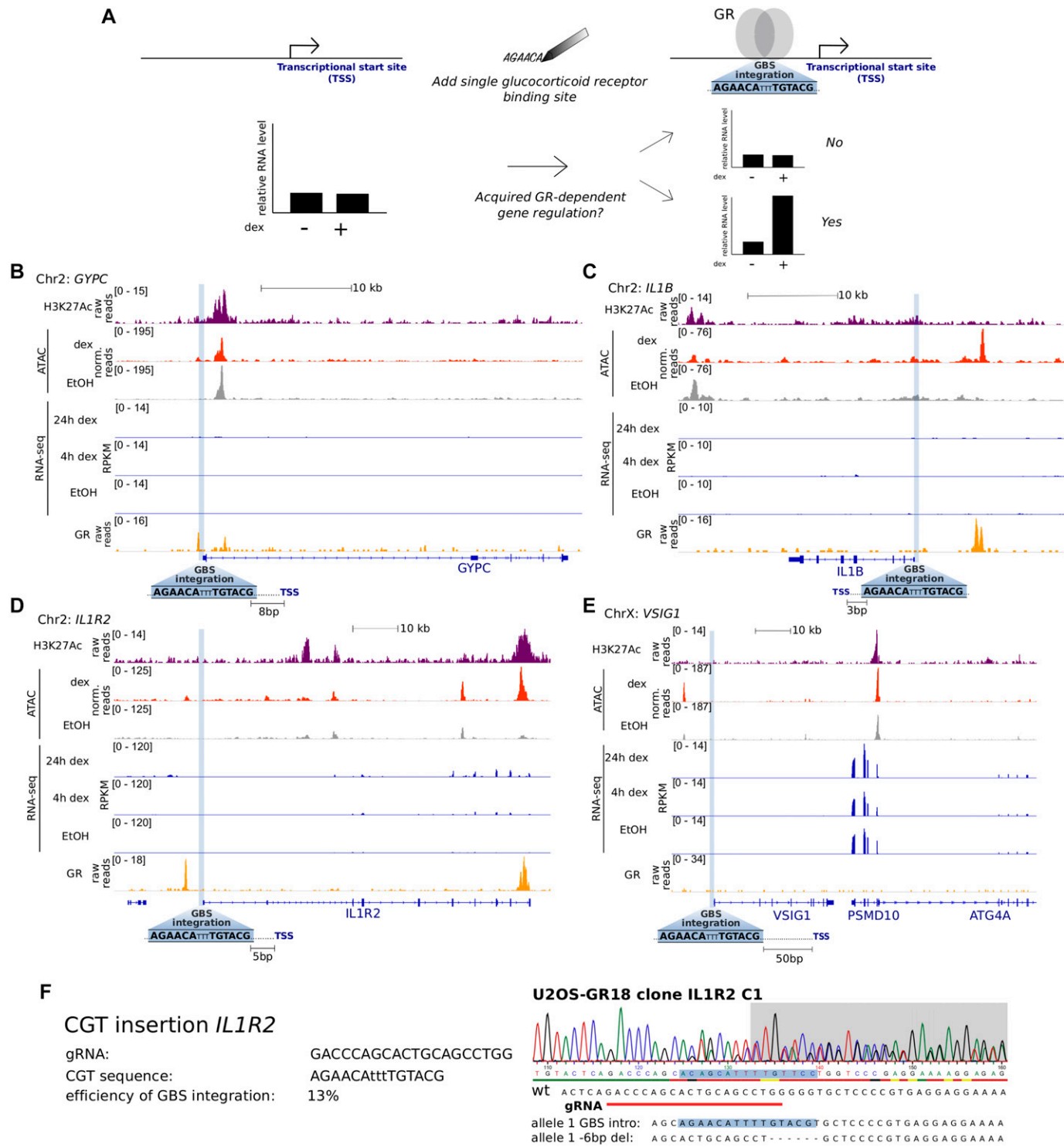

Figure 1. Genes selected for genomic GBS integration at the respective promoter region.
(A) Overview of the experimental design of our study. (B–E) Tracks showing H3K27ac and GR ChIP-seq normalized tag density, ATAC-seq, and RNA-seq reads for U2OS-GR cells (the non-edited parental cell line) that were treated as indicated. Genomic regions surrounding the loci of GBS integration are shown for *GYPC* (B), *IL1B* (C), *IL1R2* (D), and *VSIG1* (E). The genomic site targeted for GBS integration is highlighted in blue and its distance in base pairs to the TSS is indicated. (F) HDR-mediated genome editing to introduce the CGT GBS upstream of the *IL1R2* gene. The sequence of the gRNA, the sequence of the introduced GBS, and the efficiency of successfully edited single-cell–derived clonal lines are shown on the left. Sanger sequencing for a successfully edited clone and the sequence for each allele are shown on the right.

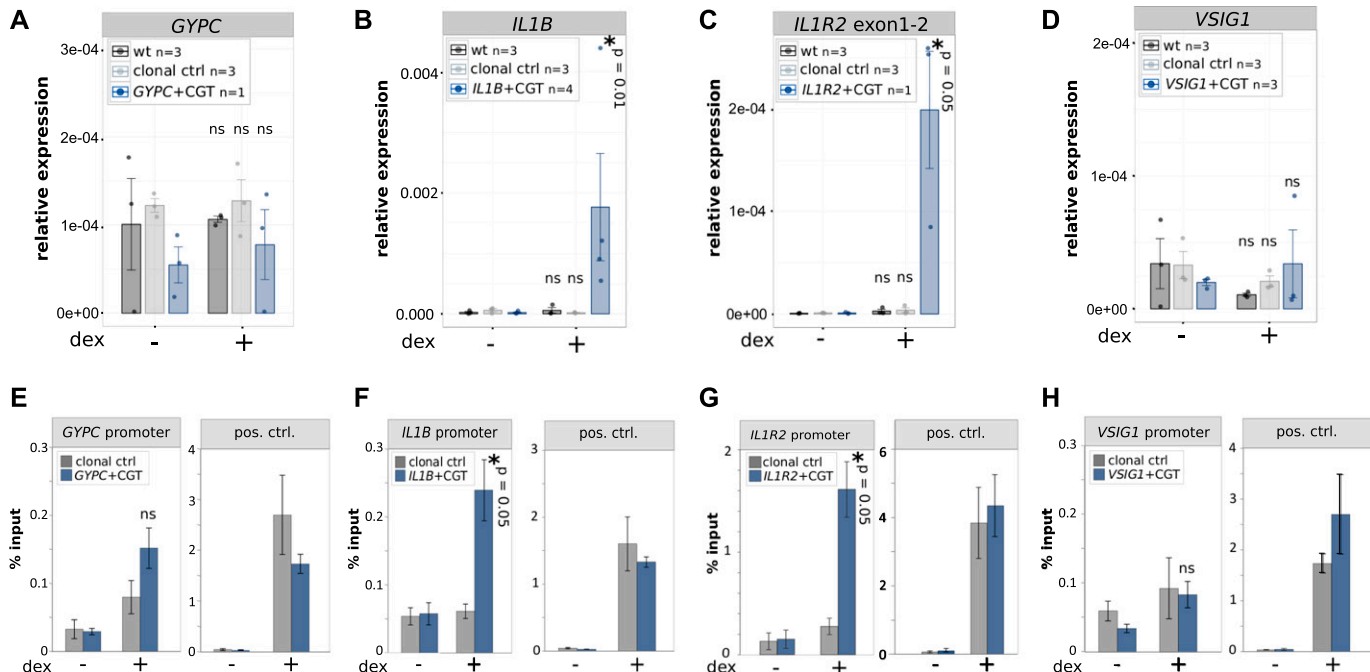

**Figure 2. Genomic insertion of a single GBS results in gene-specific acquired GR binding and GR-dependent transcriptional regulation.**
**(A–D)** Relative mRNA expression levels as determined by qPCR for (A) *GYPC* (only one single-cell–derived clonal line was analyzed: n = 1) (B) *IL1B* (n = 4), (C) *IL1R2* (n = 3), and (D) *VSIG1* (n = 3) are shown for unedited parental U2OS-GR cells (wt), for unedited clonal control cell lines, and for clonal cell lines with an integrated GBS at the target gene as indicated. Averages ± SEM for cell lines treated overnight with 1 μM dexamethasone (dex) or with ethanol (−) as vehicle control are shown. Dots show the values for each individual clonal line. Statistical tests were performed using an unpaired one-sided Mann–Whitney *U* test comparing each dex-treated group with its untreated counterpart. **(E–H)** GR occupancy at the edited genes was analyzed by ChIP followed by qPCR for cells as indicated treated with vehicle control (−) or 1 μM dex for 90 min. **(E–H)** Average percentage of input precipitated ± SEM from three independent experiments is shown for an unedited clonal control cell line and for a clonal cell line edited at either the (E) *GYPC*, (F) *IL1B*, (G) *IL1R2*, or (H) *VSIG1* locus. Left panel shows binding at the edited promoter. Right panel binding at the unedited *GILZ* locus, which serves as control for comparable ChIP efficiencies between clonal lines. Statistical tests were performed using an unpaired one-sided Mann–Whitney *U* test comparing GR binding at the *IL1R2* promoter between dex-treated clonal control and the dex-treated edited clonal line as indicated.

*VSIG1* gene, no GR recruitment was observed regardless of whether a GBS was added or not (Fig 2H). Importantly, for all clonal lines analyzed, we observed robust hormone-dependent GR binding at the endogenous GILZ locus, which served as a positive control (pos. ctrl) and shows that the ChIP efficiency was comparable between our clonal lines (Fig 2E–H).

Given GR's preference for binding at accessible chromatin (7), differences in accessibility could explain the locus-specific binding of GR to the added GBS. To test this hypothesis, we generated ATAC-seq (assay for transposase-accessible chromatin, (22)) data for parental U2OS-GR cells both in the presence and absence of dexamethasone. Visual inspection of the ATAC-seq data revealed a similar, relatively low, ATAC-seq signal (Fig 1B–E) for each of the loci examined. This indicates that the regions where we added the GBS are relatively inaccessible and that the gene-specific binding does not appear to be a consequence of marked differences in chromatin accessibility between regions. Similarly, H3K27ac levels, a marker of active enhancers, were similarly low in untreated cells for each of the genes analyzed (Fig 1B–E).

To test if activation by another transcriptional activator shows the same gene-specific activation pattern as GR, we targeted dCas9-SAM (17) to the TSS of each of the four candidate genes. The dCas9-SAM system is a powerful tool to activate genes upon TSS-proximal recruitment and consists of a nucleolytically dead Cas9 protein fused to the VP64 activation domain. In addition, the system uses a

modified guide RNA containing two MS2 stem loops that recruit MS2-p65-HSF1 fusion proteins to further boost activation. Targeting of dCas9-SAM to the TSS of the *IL1R2* and *IL1B* genes resulted in a robust activation (>100-fold increase over control non-targeting guide RNA, Fig 3). In contrast, targeting dCas9-SAM to the TSS of genes that did not acquire GR-dependent activation upon GBS addition resulted in marginal activation (2.9-fold) for *VSIG1*, whereas the *GYPC* gene was not activated by dCas9-SAM (Fig 3).

Together, our results show a similar pattern of activation by GR and by targeting the dCas9-SAM system that cannot be explained by obvious differences in chromatin accessibility based on ATAC-seq data. For GR, this gene-specific activation can be partially explained by gene-specific GR recruitment. Specifically, the most robust GR recruitment was observed for the two genes that acquired GR-dependent regulation upon GBS addition, whereas no binding was detected for *VSIG1*, which could not be converted into a GR target.

### Activation of the endogenous *IL1R2* gene by GBS variants

GR is known to bind directly to a broad spectrum of sequences that differ in their precise sequence composition (23). In addition to recruiting GR to defined genomic loci, the sequence of the binding site can also modulate the activity of GR downstream of binding (10, 11, 23). To test if GBS variants, other than the CGT variant we initially tested, can accommodate GR-dependent regulation of the *IL1R2*

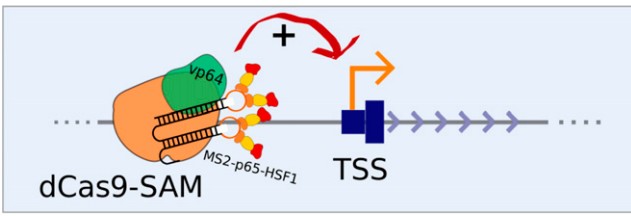

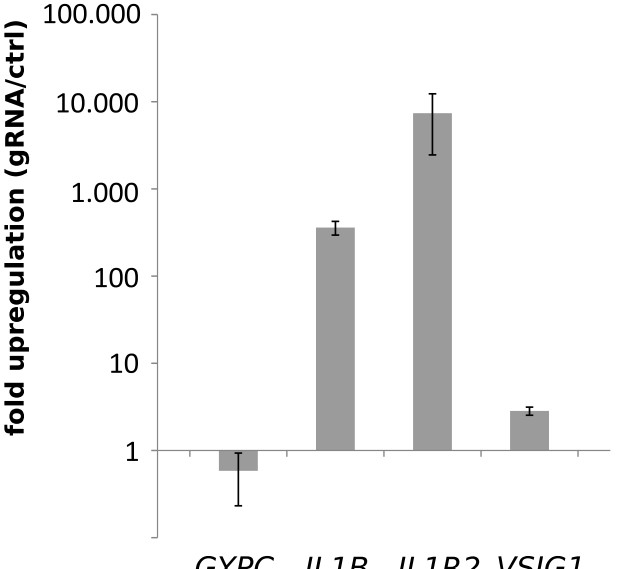

**Figure 3. Activation of targeted loci by the Cas9 activator dCas9-SAM.**
(Top) Schematic of the dCas9 synergistic activation mediator (dCas9-SAM) targeted to the promoter region of a gene. (Bottom) Fold induction of *GYPC*, *IL1B*, *IL1R2*, and *VSIG1* expression upon targeting dCas9-SAM to its TSS. The average fold change induced by a gRNA targeted to the promoter region of the respective gene relative to a control non-targeting gRNAs ± SEM from three independent experiments is shown.

gene, we generated single-cell–derived clonal lines for three additional GBS variants (Figs 4A and S2). We picked two GBS variants PAL (a synthetic sequence with perfect palindromic half sites) and GILZ (a GBS derived from a GR-bound region near the *GILZ* gene) that showed markedly lower activities than the CGT sequence in previous studies using reporter assays and one with comparable activity FKBP5-2 (a GBS derived from a GR-bound region near the *FKBP5* gene) (10, 11). Next, we tested if these GBS variants could also convert the *IL1R2* gene into a GR target and found that this was the case for each of the variants tested (Fig 4B). Accordingly, ChIP analysis showed GR binding at the TSS of the *IL1R2* gene upon the addition of each GBS variant tested (Fig 4C).

For a quantitative comparison between the GBS variants, we generated multiple independent clonal lines for each GBS variant (n ≥ 3). This is important for a meaningful comparison between GBS variants, given the high degree of variability in the level of activation observed between individual clones with the same GBS variant (Fig 2C). However, when we averaged the level of activation across clonal lines with the same GBS, we observed no significant differences between the GBS variants in the levels of *IL1R2* activation (Fig 4B). As expected, this was also the case for the endogenous target gene *DUSP1*, a GR target gene located on another

chromosome, which served as an internal control to ensure that GR activity was comparable across clonal lines (Fig 4B). Similarly, the kinetics of *IL1R2* activation after GR activation by dexamethasone was comparable for each of the GBS variants (Figs 4D and S3A for the unedited control gene *DUSP1*). Taken together, these results indicate that several GBS variants can convert the *IL1R2* gene into a GR target with similar levels of activation for each variant.

## Comparison of engineered GBS variants at the endogenous *GILZ* enhancer

We previously showed that deletion of an individual endogenous GBS (GBS1, Fig 5A) resulted in a partial reduction (~60%) of hormone-induced *GILZ* levels when compared with either parental U2OS-GR cells or to unedited clonal controls ((13) and Fig 5B). To test if other GBS variants can substitute for the endogenous GBS1, we first analyzed the activity of several variants using a luciferase reporter (Fig 5C and D). We found that the level of activation was comparable for different GBS variants with the exception of the PAL sequence, a high-affinity GBS variant (10), which showed a lower level of activation (Fig 5D). Based on these findings, we decided to use HDR to convert the endogenous GBS1 to the FKBP5-2 or to the PAL sequence (Fig S4), the two variants with the highest and lowest reporter activity, respectively. Next, we analyzed the effect of changing the sequence of the endogenous GBS1 using HDR and found that the level of activation for the PAL and FKBP5-2 variants was markedly higher than that observed when the GBS1 was deleted (Fig 5B and E). In fact, the activation for both variants was indistinguishable from the activation observed for parental U2OS-GR cells or for unedited clonal controls (Fig 5E). This was also observed for the unedited *DUSP1* gene, which serves as a control to make sure that GR activity is comparable among the clonal lines we analyzed (Fig 5E). Notably, the in vitro affinity of GR for the PAL sequence is an order of magnitude higher than for the endogenous GBS1 sequence (10). To test if the higher affinity of GR for the PAL sequence might facilitate *GILZ* activation at lower hormone concentrations, we assayed the levels of activation observed at 100 pM and 10 nM dexamethasone. Consistent with our expectation, activation of the *GILZ* gene was lower at lower hormone concentrations (Fig 5F). However, when comparing GBS variants at a given hormone concentration, we observed similar levels of activation, indicating that the sequence identity of the GR-binding site does not affect the dose response of the *GILZ* gene (Figs 5F and S5A for the unedited control gene *FKBP5*). Similarly, the kinetics of activation after hormone treatment was comparable for each of the GBS variants analyzed (Figs 5G and S5B for the unedited control gene *DUSP1*). Together, these results indicate that both the PAL and the FKBP5-2 GBS variants can substitute for the original GBS1 sequence at the *GILZ* gene without apparent differences between GBS variants in the level of GR-dependent activation observed.

## Discussion

The ultimate quest in deciphering *cis*-regulatory logic is to reach a level of understanding that would allow an accurate quantitative and

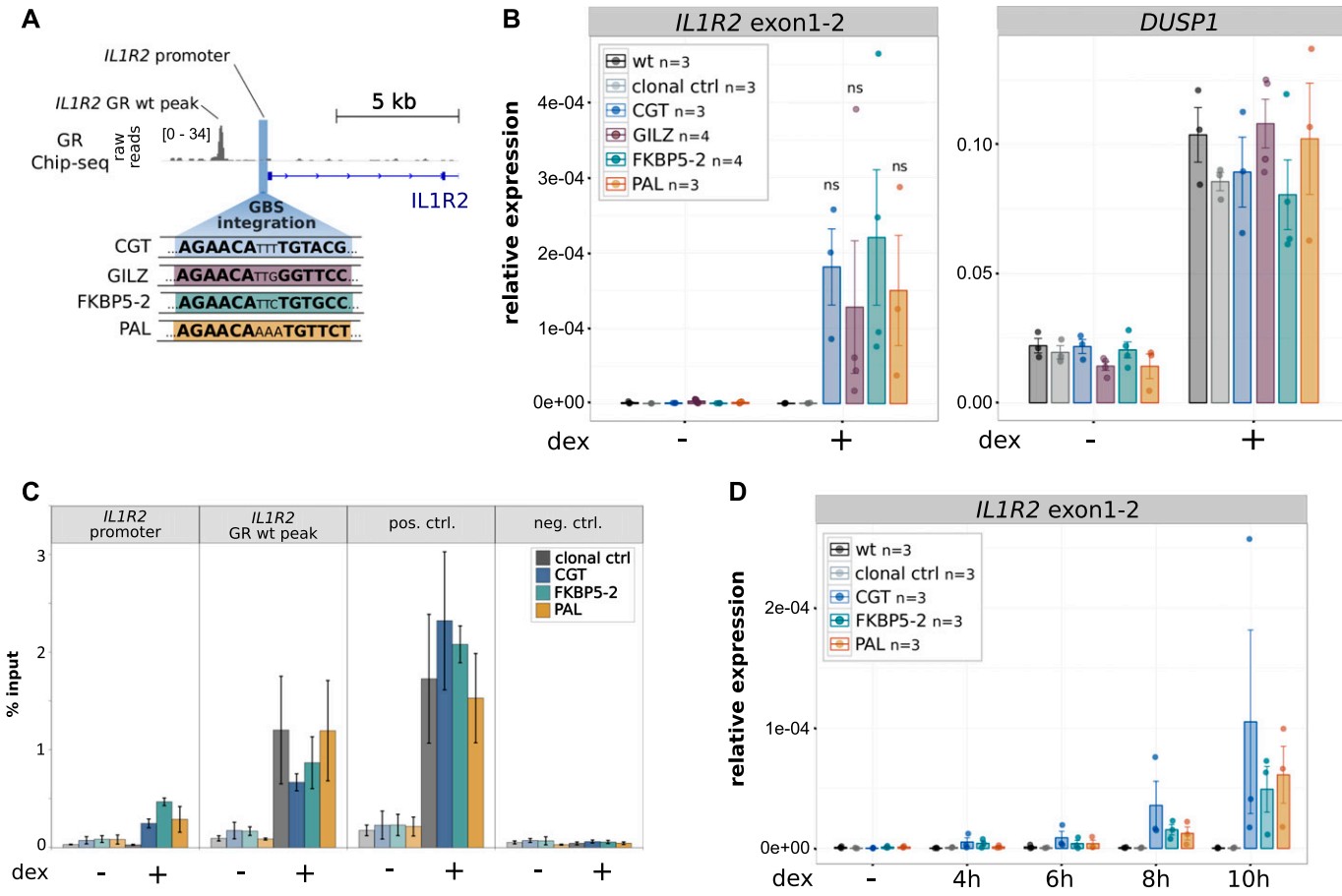

**Figure 4. Comparison of IL1R2 activation levels by inserted GBS variants.**
**(A)** Overview of the *IL1R2* promoter region showing the location of the GBS integration, the sequence of integrated GBS variants, the GR ChIP-seq tag density for dex-treated U2OS-GR cells and the location of a GR ChIP-seq peak that is already present at the locus before editing (*IL1R2* GR wt peak). **(B)** Relative mRNA expression levels as determined by qPCR for *IL1R2* and for the unedited control GR target gene *DUSP1* are shown for unedited parental U2OS-GR cells (wt), for unedited clonal control cell lines, and for clonal cell lines with an integrated GBS as indicated at the *IL1R2* gene. Averages ± SEM for cell lines treated overnight with 1 μM dexamethasone (dex) or with ethanol (–) as vehicle control are shown. Dots show the values for each individual clonal line. Statistical tests were performed using an unpaired two-sided Mann–Whitney *U* test comparing the RNA level for dex-treated FKBP5-2 GBS samples with the dex-treated RNA levels for each of the other GBS variants analyzed. **(C)** GR occupancy was analyzed by ChIP followed by qPCR for clonal lines as indicated and treated with vehicle control (–) or 1 μM dex for 90 min. Average percentage of input precipitated ± SEM from three independent experiments is shown for the locus where the GBS was inserted (*IL1R2* promoter), the *IL1R2* wt peak, a positive control region (*GILZ*), and a negative control region (*TAT*). **(D)** Relative mRNA expression levels as determined by qPCR for the *IL1R2* gene for unedited parental U2OS-GR cells (wt), for unedited clonal control cell lines and for clonal cell lines with an integrated GBS as indicated at the *IL1R2* gene. Averages ± SEM for cell lines treated for 4, 6, 8, or 10 h with 1 μM dex or vehicle control (–) is shown. Dots indicate the value of each individual clonal cell line.

qualitative prediction of gene expression levels based on regulatory sequence composition. If identified, such knowledge would, for example, facilitate the rational design of gene regulatory circuits to generate cells with desired gene expression patterns. A broad range of methods can help identify the rules that govern transcriptional output, including perturbation experiments, genome-wide mapping of functional elements and high-throughput reporter assays to measure the activity of promoter and enhancer regions (24). By building synthetic circuits, the accuracy of the elusive regulatory code can be tested and can help identify possible gaps in our knowledge when the predicted results are in conflict with the measured transcriptional output. Here, we used genome editing to evaluate two traits associated with GR-dependent gene regulation.

The first trait we evaluated is the positive correlation between GR-dependent regulation of a gene and promoter-proximal GR

binding (13). By adding a promoter-proximal GBS to several genes, we could demonstrate that the addition of a single GR-binding site can be sufficient to convert a gene normally not regulated by GR into a target gene. For the converted genes (*IL1B* and *IL1R2*, Fig 2), we observed robust GR recruitment to the added GBS. In contrast, either no recruitment or a less robust acquired recruitment of GR was found for the genes that could not be converted (*VSIG1* and *GYPC*, Fig 2) indicating that GR binding is required for the acquired GR-dependent regulation. Interestingly, the equivalent gene-specific ability to activate genes was found for the synthetic dCas9-SAM activator. Because GR almost exclusively binds to regions of open chromatin (7), differences in chromatin accessibility would provide a straightforward explanation for the gene-specific acquired activation observed. However, arguing against this explanation, we did not observe obvious differences in chromatin

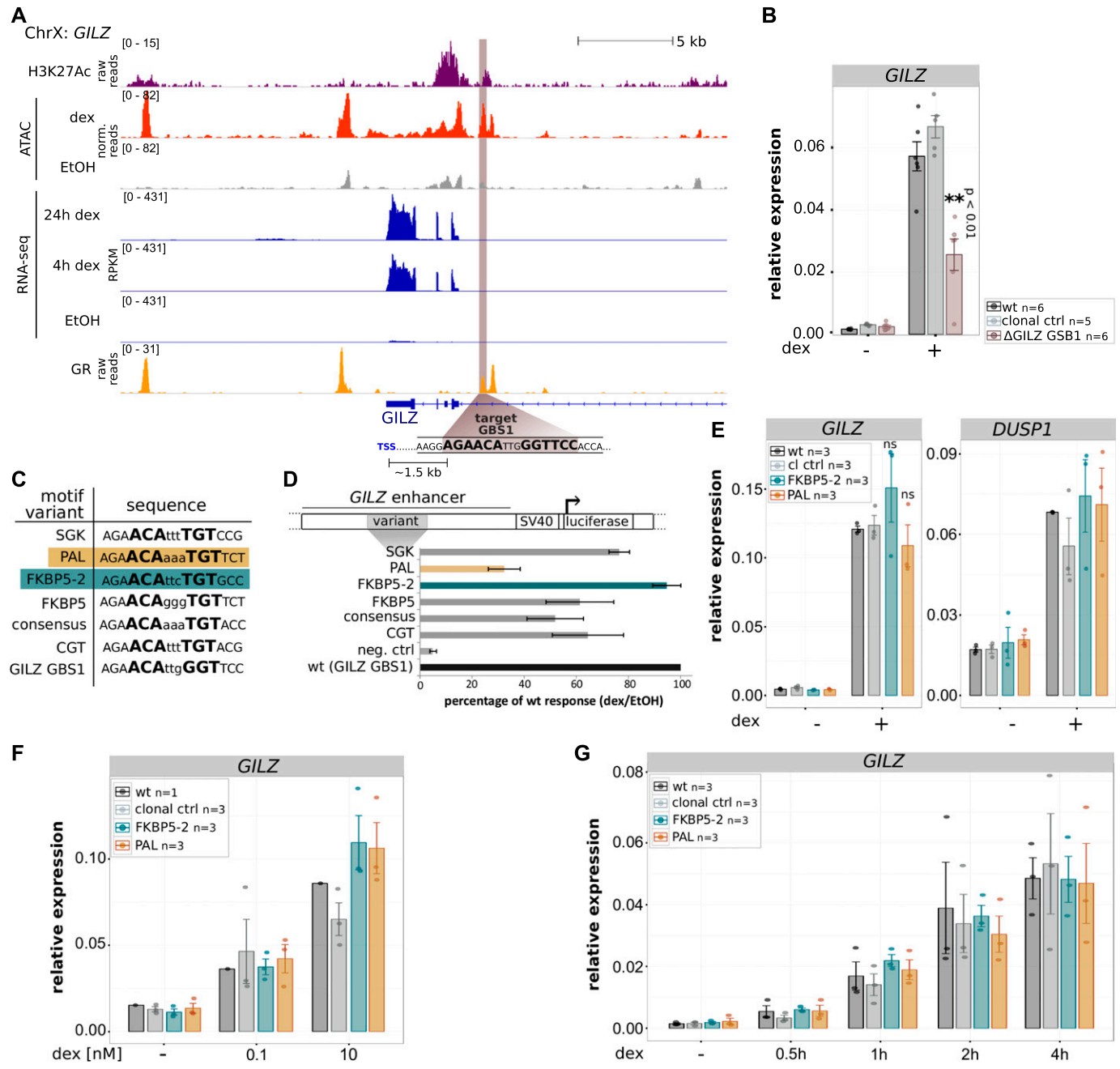

**Figure 5. Effect of GBS1 sequence identity on GR-dependent *GILZ* activation.**
**(A)** Tracks showing H3K27ac and GR ChIP-seq tag density, ATAC-seq, and RNA-seq reads at the *GILZ* locus for U2OS-GR cells treated as indicated. The *GILZ* GBS1 targeted for editing is highlighted in brown and the distance in kilo base pairs (kb) to the next TSS is indicated. **(B)** Relative *GILZ* mRNA expression is shown for parental U2OS-GR cells, for unedited clonal controls, and for clonal lines with a deleted GBS1. The average ± SEM of at least five clonal cell lines treated overnight with 1 μM dex or vehicle control (–) is shown. Dots show the values for each individual clonal line. Statistical tests were performed using an unpaired two-sided Mann–Whitney *U* test comparing dex-treated *GILZ* levels between clonal lines with a deleted GBS1 and their unedited clonal control counterpart. **(C)** DNA sequence of GBS variants analyzed. **(D)** Relative fold activation of luciferase reporters with GBS variants as indicated comparing cells treated with vehicle control (etoh) and cells treated overnight with 1 μM dex. Averages ± SEM from three independent experiments are shown. **(E)** Relative mRNA expression levels as determined by qPCR for *GILZ* and for the unedited control GR target gene *DUSP1* are shown for unedited parental U2OS-GR cells (wt), for unedited clonal control cell lines, and for clonal cell lines with GBS variant as indicated at the *GILZ* GBS1 locus. Averages ± SEM for cell lines treated overnight with 1 μM dexamethasone (dex) or with ethanol (–) as vehicle control are shown. Dots show the values for each individual clonal line. Statistical tests were performed using an unpaired two-sided Mann–Whitney *U* test comparing dex-treated *GILZ* levels between clonal lines for each introduced GBS variant and their unedited clonal control counterpart. **(E, F)** Same as for (E) except that *GILZ* mRNA levels are shown for cells treated overnight with 0.1 nM dex, 10 nM dex, or vehicle control (–). **(E, G)** Same as for (E) except that cells were treated for 0.5, 1, 2, or 4 h with 1 μM dex.

accessibility or differences in H3K27ac levels between converted genes and genes that could not be converted into GR targets for unedited parental U2OS-GR cells (Fig 1). However, we cannot rule out that the GBS introduction might influence H3K27ac levels, DNA accessibility, or other chromatin characteristics at the edited locus, although some of our findings argue against profound changes. First, introduction of the GBS does not influence basal expression levels of the edited genes (Fig 2). Second, the activation by targeted recruitment of dCas9-SAM, which was measured in non-manipulated parental cells, follows the same gene-specific pattern of activation that was observed for GR. Finally, ChIP experiments indicate that H3K27ac levels at the edited *IL1R2* promoter locus do not change upon addition of a GBS (Fig S3B) either in the presence or absence of GR activation by the addition of dexamethasone. The presence of endogenous GR-binding sites near the *IL1B* and *IL1R2* genes (Fig 1C and D) and the absence of a GR-binding site near the *VSIG1* gene could also explain why the addition of a GBS is able to convert only *IL1B* and *IL1R2* into GR targets. However, the presence of a nearby GR-binding site does not appear to be the entire explanation for gene-specific acquired gene regulation, given that the *GYPC* gene, which cannot be converted, also harbors GR-binding sites near its TSS (Fig 1B). Alternative explanations for the gene-specific acquirement of activation could be differences in DNA methylation and that the sequence context for the *IL1R2* and *IL1B* encodes recognition sequences for factors that accommodate GR binding and activation from the introduced GBS. Sequence features that accommodate GR-dependent activation could be identified computationally. However, this would require the editing of a larger

number of genes to find common features among converted genes. Alternatively, we could disrupt candidate sequences present at converted genes to assay their role in accommodating GR-dependent activation. Notably, the initial response element we introduced consisted of just a single GBS sequence, which to our surprise was sufficient to convert some genes into GR targets. Likely, activation of other genes requires more complex response elements consisting of multiple GBS's or of a GBS and binding sites for TFs known to synergize with GR (25). In addition, the ability of a single GBS to convert a gene into a GR target might be cell type–specific, something we intend to test in the future.

Profound changes in GR occupancy patterns are also observed when GR binding is compared between mouse and human macrophages (26). This divergence is accompanied by changes in the repertoire of responsive genes between species and is associated with gains and losses of GR recognition sequences (26). Similar to the targeted nucleotide substitutions we introduced here, the evolutionary turnover of GR-binding sites is predominantly driven by nucleotide substitutions as a consequence of mutations (26). Notably, in contrast to the promoter-proximal GBSs we added, most of the endogenous GR binding occurs promoter-distal (4, 7, 12). In fact, GR binding is biased against accessible chromatin located at promoter regions, which can be partially explained by the presence of fewer GR recognition sequences in promoter regions when compared with their promoter-distal counterparts (4, 12). A possible reason for this bias is that there might be selection against promoter-proximal GR binding to safe-guard cell type–specific transcriptional consequences of glucocorticoid signaling given that

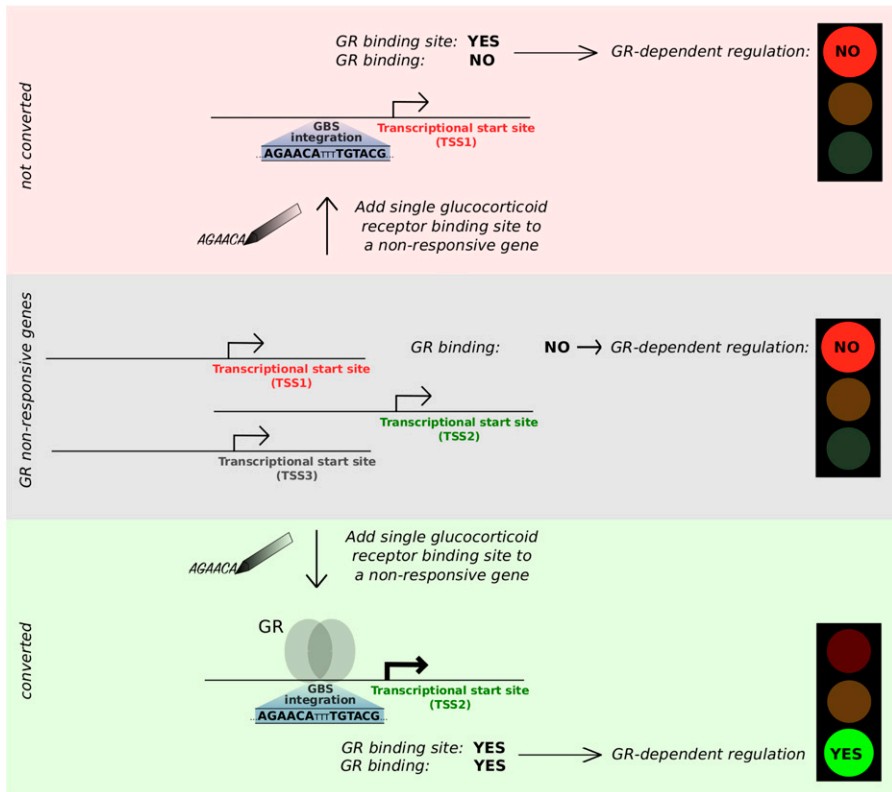

**Figure 6.  Unsophisticated enhancer logic: addition of a single occupied GBS suffices to convert a nonresponsive gene into a GR target.**

promoter-proximal GR binding is associated with gene regulation regardless of cell type examined, whereas distal binding is more likely to result in cell type–specific gene regulation ([4]).

The second trait we evaluated is the sequence of GR's DNA-binding site, which can influence the magnitude of transcriptional activation by GR ([10], [11], [27]). By changing the sequence identity of the introduced GR-binding site at the *IL1R2* gene and of a GBS at an endogenous GR-bound region near the *GILZ* gene, we found that distinct variants facilitate equivalent levels of GR-dependent activation. This indicates that the sequence identity of the GBS does not direct markedly distinct levels of activation for either locus examined. Our findings in the genomic context are in contrast to previous studies showing that the sequence identity of the GBS can have an impact on the magnitude of GR-dependent activation ([10], [11]). One possible explanation for this discrepancy is that our approach may not have the sensitivity to detect subtle differences in activity. For example, because of the high levels of variation in the level of activation we observe for a given GBS variant when comparing individual clonal lines (Fig 4C). This clonal variability precludes the identification of significant differences when small numbers of clonal lines are analyzed. Another possible explanation is that the ability of GBS variants to induced different levels of activation is context specific. Notably, the expression of endogenous GR target genes can be controlled by one or multiple GR-bound enhancers ([13], [28]), which might mask GBS-specific activities. Furthermore, studies showing GBS-specific activities ([10], [27], [29]) were performed using minimal promoters (SV40 or thymidine kinase), whereas we studied activation from the endogenous promoters of the *GILZ* and *IL1R2* genes, respectively. Thus, GBS-specific levels of activation might only occur for specific types of promoters, an idea we would like to pursue in forthcoming studies.

Taken together, our engineering of *cis*-regulatory elements argue for an unsophisticated enhancer logic ([30]) where a single occupied GBS can be sufficient to activate genes when it is located promoter-proximal (Fig 6). Moreover, we find that acquired activation can be mediated by distinct GBS variants without obvious differences in activity between variants. To our knowledge, this is the first study to use genome engineering to add a binding site for a mammalian TF to expand its repertoire of endogenous target genes. We started by building response elements consisting of just a single GR-binding site. By adding complexity to the response element, increasing the distance between the response element and the promoter and by analyzing larger numbers of genes, the engineering approach provides a framework to refine our understanding of the *cis*-regulatory logic of gene regulation which would ultimately facilitate the construction of cells with desired gene expression profiles.

# Materials and Methods

## Cell culture, transient transfections, and luciferase assays

U2OS cells with stably integrated rat GR α ([21]) were cultured in DMEM supplemented with 5% FBS at 37°C and 5% $CO_2$. The pGILZ1 construct containing a GR-bound region near the *GILZ* gene driving a luciferase reporter gene has been described previously ([31]). GBS1, encoded in the pGILZ1 construct, was mutated by site-directed mutagenesis using primers listed in Table 1. Transient transfections of U2OS-GR cells were performed as described previously ([10]). Luciferase activity was measured using the Dual Luciferase Reporter Assay kit (Promega).

## RNA-seq U2OS-GR cells

RNA-seq data for U2OS-GR cells was generated as previously described ([32]), except that cells were treated with 1 μM dexamethasone for 24 h in addition to the 4-h treatment.

## ATAC-seq U2OS-GR cells

For ATAC-seq, U2OS-GR cells were treated with 1 μM dexamethasone or vehicle control (ethanol) for 90 min. ATAC-seq was performed with 100,000 cells per treatment according to the Omni-ATAC-seq protocol ([33]), with the following modifications: (1) the transposase reaction was stopped precisely after 30 min through the addition of 2.5 μl of 10% SDS and (2) the transposed DNA fragments were PCR-amplified using the p5-containing primer 5′–AATGATACGGCGACCACCGAGATCTACACTCGTCGGCAGCGTC-3′ and the p7-containing primer 5′-CAAGCAGAAGACGGCATACGAGATGTA-AGTCACGTCTCGTGGGCTCGG-3′ or 5′-CAAGCAGAAGACGGCATACGAG-ATTTCAGTGAGGTCTCGTGGGCTCGG-3′ (p7-containing primers have different barcodes for multiplexed sequencing).

Libraries were sequenced on an Illumina HiSeq 4000 with 50-bp paired-end reads to a sequencing depth of 50 M reads. Raw reads were mapped to the reference assembly hg19 using Bowtie2 v.2.1.0 (–very-sensitive) ([34]). SAMtools ([35]) was used for conversion of SAM

**Table 1. Primer sequences for SDM of luciferase reporter constructs and HDR templates.**

| Name | Sequence 5′ to 3′ |
| --- | --- |
| *GILZ* wt GBS to FKBP5-2 | CAGGACCAAAGG**AGAACATCCTGTGCC**ACCACATATAC |
| | GTATATGTGGTGGCACAGGATGTTCTCCTTTGGTCCTG |
| *GILZ* wt GBS to PAL | CAGGACCAAAGG**AGAACAAAATGTTC**TACCACATATAC |
| | GTATATGTGGTAGAACATTTTGTTCTCCTTTGGTCCTG |
| *IL1R2* intro CGT GBS | TACTCAGACCCAGC**AGAACATTTTGTACG**TGCTCCCCGTGAG |
| | CTCACGGGGAGCACGTACAAAATGTTCTGCTGGGTCTGAGTA |
| *IL1R2* intro PAL GBS | TACTCAGACCC**AGCAGAACAAAATGTTCT**TGCTCCCCGTGAG |
| | CTCACGGGGAGCAAGAACATTTTGTTCTGCTGGGTCTGAGTA |
| *IL1R2* intro FKBP5-2 GBS | TACTCAGACCCAGC**AGAACATCCTGTGCC**TGCTCCCCGTGAG |
| | CTCACGGGGAGCAGGCACAGGATGTTCTGCTGGGTCTGAGTA |
| *IL1R2* intro GILZ GBS | TACTCAGACCCAGC**AGAACATTGGGTTCC**TGCTCCCCGTGAG |
| | CTCACGGGGAGCAGGAACCCAATGTTCTGCTGGGTCTGAGTA |
| *GYPC* intro CGT GBS | AATTCTCAACC**AGAACATTTTGTACG**GGTAG |
| | CTACCCGTACAAAATGTTCTGGTTGAGAATT |
| *IL1B* intro CGT GBS | GGTTTGGTATC**AGAACATTTTGTACG**CGCTG |
| | CAGCGCGTACAAAATGTTCTGATACCAAACC |
| *VSIG1* intro CGT GBS | TTATTAACACAGTA**AGAACATTTTGTACG**AAACACGCC |
| | GGCGTGTTTCGTACAAAATGTTCTTACTGTGTTAATAA |

**Table 2. Primer sequences for the quantification of GR binding in ChIP experiments.**

| Name | Sequence 5' to 3' |
|---|---|
| IL1R2 promoter | AAAAATAGGGAAACTTATGCGGC |
| | ACCTTTTCCTCCTCACGGG |
| IL1R2 wt GR-peak | TGCAATAAACATCCTGGGTGA |
| | GTGTCCACCACCAATAGCAC |
| Positive control (GILZ) | AACTCAGCAGCTTTTCTTCGT |
| | AACCAAGGAATTGGGTCACAT |
| Negative control (TAT) | AATGGCAGCCCCTAGTCATTC |
| | AACTGGGAGTGATACTGGTTCC |

to BAM files and sorting. Duplicate reads were removed with Picard tools v.2.17.0 (MarkDuplicates) (http://broadinstitute.github.io/picard/). BigWig files were generated with bamCoverage from deepTools (36).

### ChIP and ChIP-seq

ChIP-qPCR for GR and H3K27ac (C15410196; Diagenode) was performed as previously described (13) using primers listed in Tables 2 and 3. For H3K27ac ChIP-seq experiments, U2OS-GR cells were treated for 1.5 h with 1 µM dexamethasone, cross-linked with 1% formaldehyde for 3 min, and harvested. Chromatin was precipitated using 1 µg of anti-H3K27ac antibody (C15410196; Diagenode). Sequencing libraries were prepared using the NEBNext Ultra DNA Library Prep kit (E7370; NEB) according to the manufacturer's instructions and submitted for paired-end Illumina sequencing. Data processing: paired-end Illumina sequencing reads were mapped to the human genome (hg19) using STAR (–alignIntronMax 1) (37) and converted to the bigWig format for visualization.

### Genome editing

The HDR templates for genome editing were generated by cloning an ~2-kb genomic region flanking the targeted integration site (genomic coordinates of cloned regions listed in Table 4) into the zero blunt PCR cloning vector (Thermo Fisher Scientific). Sequence changes in

**Table 3. Primer sequences for the quantification of H3K27Ac in ChIP experiments.**

| Name | Sequence 5' to 3' |
|---|---|
| IL1R2 promoter | AAAAATAGGGAAACTTATGCGGC |
| | ACCTTTTCCTCCTCACGGG |
| IL1R2 wt GR-peak | TGCAATAAACATCCTGGGTGA |
| | GTGTCCACCACCAATAGCAC |
| Positive control (SYN2) | AGGAATATTTGCTGACACTTCCA |
| | ACAGCACCTACCATATAGGCTT |
| Negative control (TAT) | AATGGCAGCCCCTAGTCATTC |
| | AACTGGGAGTGATACTGGTTCC |

**Table 4. Genomic region spanned by HDR templates.**

| Gene | Location (GRCh37/hg19) |
|---|---|
| GILZ | ChrX:106,960,177-106,962,953 |
| IL1R2 | Chr2:102,606,778-102,609,287 |
| GYPC | Chr2:127,412,437-127,414,329 |
| IL1B | Chr2:113,593,542-113,595,924 |
| VSIG1 | ChrX:107,287,118-107,289,154 |

these templates were introduced by site-directed mutagenesis, using the primers listed in Table 1. To avoid repeated Cas9-editing modifications, the added GBSs overlapped with the gRNA target sequence (genomic location of introduced GBSs listed in Table 5). gRNAs for genome editing (Table 6) were designed using the CRISPOR webtool (http://crispor.tefor.net/) and cloned into the sgRNA/Cas9 expression construct PX459 (#62988; Addgene). To generate clonal lines with HDR-induced sequence changes, U2OS-GR cells were transfected using 600 ng of the gRNA construct and 3 µg of the HDR template by nucleofection (Lonza Nucleofector kit V) according to the manufacturer's instructions. Subsequently, successfully transfected cells were selected by treating cells with puromycin (2.5 µg/ml) for 24 h. To increase gene editing by HDR, we treated transfected cells for 24 h with 10 µM SCR7 (XcessBio Biosciences). Single-cell–derived clonal cell lines were genotyped by PCR using genomic DNA isolated with the DNeasy Blood and Tissue kit (QIAGEN) and primers binding outside the HDR template.

### RNA preparation and analysis by quantitative real-time PCR (qPCR)

Cells were cultured to confluency and treated with dexamethasone or vehicle control (ethanol) for the times and hormone concentrations as indicated in the figure legends. After the hormone treatment, RNA was extracted, reverse-transcribed, and analyzed by qPCR as described previously (10) using the primer pairs listed in Table 7. For the analysis of lowly expressed genes (IL1R2, VSIG1, IL1B, and GYPC), the cDNA was diluted 1:3.5; for all other genes 1:25.

### dCas9-SAM activation of endogenous genes

To test if dCas9-SAM could activate endogenous target genes when recruited to the sites where we added the GBSs, we created gRNAs containing MS2 loops by cloning the target sequence (Table 6) into the sgRNA(MS2) plasmid (#61424; Addgene). Next, U2OS-GR cells were transfected with 600 ng each of the MS2-containing gRNA,

**Table 5. Location of GBS introduction.**

| Gene | Location (GRCh37/hg19) |
|---|---|
| GILZ | ChrX:106,961,576-106,961,591 |
| IL1R2 | Chr2:102,608,286-102,608,301 |
| GYPC | Chr2:127,413,491-127,413,506 |
| IL1B | Chr2:113,594,360-113,594,375 |
| VSIG1 | ChrX:107,288,135-107,288,150 |

**Table 6. gRNA sequences for gene editing and activation by dCas9-SAM.**

| Name | Sequence 5′ to 3′ | PAM |
|---|---|---|
| GILZ | CAGGACCAAAGGAGAACATT | GGG |
| IL1R2 | GACCCAGCACTGCAGCCTGG | GGG |
| GYPC | TCAACCACAACCTCTGTATC | CGG |
| IL1B | GAAAGCCATAAAAACAGCGA | GGG |
| VSIG1 | ACACAGTAGCAAATATATCA | AGG |

dCas9-VP64 expression construct (#48223; Addgene), and an MS2-p65-HSF1 activator expression construct (#61423; Addgene) by nucleofection (Lonza Nucleofector kit V) according to the manufacturer's instructions. 24 h after transfection, total RNA was isolated using the RNeasy kit (QIAGEN), DNase-I digested and reverse transcribed using random primers (NEB) and analyzed by qPCR using primers listed in Table 7. To calculate the fold up-regulation, we first calculated the RNA level of the targeted gene relative to *RPL19* (which served as an internal control) observed when cells were transfected with the promoter-targeting gRNA. In addition, we determined the RNA level of the targeted gene for each of three guide RNAs that target the promoter of another gene. Finally, the fold regulation by the targeting gRNA is determined by dividing the RNA level for the targeting gRNA by the average RNA level for the three non-targeting gRNAs.

### Data access

Data to create the genome-browsed screenshots (Figs 1, 4, 5) can be found at ArrayExpress: ChIP-seq data GR: E-MTAB-2731;

**Table 7. Primer sequences for the quantification of gene expression.**

| Name | Sequence 5′ to 3′ |
|---|---|
| DUSP1 | CTGCCTTGATCAACGTCTCA |
| | GTCTGCCTTGTGGTTGTCCT |
| FKBP5 | TGAAGGGTTAGCGGAGCAC |
| | CTTGGCACCTTCATCAGTAGTC |
| GILZ | CCATGGACATCTTCAACAGC |
| | TTGGCTCAATCTCTCCCATC |
| IL1R2 exon 1–2 | CAGGTGAGCAGCAACAAGG |
| | TGCTCCTGACAACTTCCAGA |
| IL1R2 exon 8–9 | TTTTCAGACACTACGCACCA |
| | GATGAGGCCATAGCACAGT |
| GYPC | TCCAGGGATGTCTGGATGG |
| | CGAAGAGGAGGGAGACTAGG |
| IL1B | CCACAGACCTTCCAGGAGAATG |
| | GTGCAGTTCAGTGATCGTACAGG |
| VSIG1 | AGCCAATTTCTCACAGCTCG |
| | AAGTAAATCTCAGAGGTCCAGC |
| RPL19 | ATGTATCACAGCCTGTACCTG |
| | TTCTTGGTCTCTTCCTCCTTG |

RNA-seq data for U2OS-GR cells: E-MTAB-6738 and E-MTAB-7745; ChIP-seq data H3K27ac: E-MTAB-7747; and ATAC-seq data: E-MTAB-7746.

# Supplementary Information

# Acknowledgements

We thank Edda Einfeldt for excellent technical support. This work was supported by the Else Kröner-Fresenius-Stiftung (grant 2014_A152 to SH Meijsing and M Borschiwer).

### Author Contributions

V Thormann: conceptualization, formal analysis, investigation, visualization, methodology, writing—original draft, and project administration.
LV Glaser: investigation and writing—original draft.
MC Rothkegel: investigation and writing—original draft.
M Borschiwer: investigation and writing—original draft.
M Bothe: investigation and writing—original draft.
A Fuchs: investigation and writing—original draft.
SH Meijsing: conceptualization, formal analysis, supervision, funding acquisition, visualization, and writing—original draft.

### Conflict of Interest Statement

The authors declare that they have no conflict of interest.

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
