## [Reviewer comments · Life Science Alliance]

Life Science Alliance

Engineering genomic response elements to expand the glucocorticoid receptor target gene repertoire

Verena Thormann, Laura Glaser, Maika Rothkegel, Marina Borschiwer, Melissa Bothe, Alisa Fuchs, and Sebastiaan Meijnsing

DOI: <https://doi.org/10.26508/lsa.201800283>

Corresponding author(s): Sebastiaan Meijnsing, Max Planck Institute for Molecular Genetics, Berlin, Germany

Review Timeline:

Submission Date:	2018-12-17
Editorial Decision:	2019-01-28
Revision Received:	2019-02-19
Editorial Decision:	2019-02-27
Revision Received:	2019-03-05
Accepted:	2019-03-05

Scientific Editor: Andrea Leibfried

Transaction Report:

January 28, 2019

Re: Life Science Alliance manuscript #LSA-2018-00283

Dr. Sebastiaan H. Meijnsing
Max Planck Institute for Molecular Genetics, Berlin, Germany
Computational Molecular Biology
Innestrasse 63-73
Berlin 14195
Germany

Dear Dr. Meijnsing,

Thank you for submitting your manuscript entitled "Expanding the repertoire of glucocorticoid receptor target genes by engineering genomic response elements" to Life Science Alliance. The manuscript was assessed by expert reviewers, whose comments are appended to this letter.

As you will see, the reviewers think that your work is of value to others. They raise a few concerns that seem straightforward to address. We would thus like to invite you to submit a revised version of your work, following the constructive input provided by the reviewers.

Thank you for this interesting contribution to Life Science Alliance. We are looking forward to receiving your revised manuscript.

Sincerely,

Andrea Leibfried, PhD

Executive Editor
Life Science Alliance
Meyerhofstr. 1
69117 Heidelberg, Germany
t +49 6221 8891 502
e a.leibfried@life-science-alliance.org
www.life-science-alliance.org

- A letter addressing the reviewers' comments point by point.
- An editable version of the final text (.DOC or .DOCX) is needed for copyediting (no PDFs).
- High-resolution figure, supplementary figure and video files uploaded as individual files: See our detailed guidelines for preparing your production-ready images, <http://life-science-alliance.org/authorguide>
- Summary blurb (enter in submission system): A short text summarizing in a single sentence the study (max. 200 characters including spaces). This text is used in conjunction with the titles of papers, hence should be informative and complementary to the title and running title. It should describe the context and significance of the findings for a general readership; it should be written in the present tense and refer to the work in the third person. Author names should not be mentioned.

B. MANUSCRIPT ORGANIZATION AND FORMATTING:

Full guidelines are available on our Instructions for Authors page, <http://life-science-alliance.org/authorguide>

Reviewer #1 (Comments to the Authors (Required)):

This paper by Thormann et al studies the interesting question of whether transcription factor binding and mRNA expression of the nearby target gene are causally linked, and whether adding a transcription factor binding site to other genes can convert them into responsive targets. The authors use the U2OS cell line and Crispr-Cas9 genome engineering to add Glucocorticoid Receptor binding sites to four 'non-target' genes. The GR is a ligand-controlled transcription factor

with palindromic response elements, which together with its clinical relevance, makes for an ideal model to study this question.

To my knowledge, this is one of the first studies using Crispr-Cas9 to add binding sites to genes to determine if they can recruit the receptor and be transformed into targets. The approach is novel and of interest to a wider audience, with general applicability to other regulatory proteins.

However, the current manuscript suffers from a few shortcomings that should be addressed before publication:

- The authors find that two genes, *Il1b* and *Il1r2*, do respond and can be converted, while the other two, *Gypc* and *Vsig1*, do not. However, in Figure 1, it seems that there is already GR binding near these genes (but not at the promoter), and not at the non-responders. This could be the entire explanation for why the experiment works for those two. Please discuss (especially since this does not classify them as non-targets).

- Figure 2, A states $n=1$ for *Gypc*, but I assume it is $n=3$? E, f, g and h show the *Gilz* locus, but enrichment varies between 1.5 and 4 % input. Was this data normalized?

- Figure 3, what are the controls for the dCas9 SAM? Is there equal recruitment to each locus?

- Figure 4c, how far is the wt peak from the promoter at the *Il1r2* locus? Can the ChIP-qPCR really discriminate between the two sites? It might be helpful to normalize the data to the control, to see fold enrichment.

In general, some graphs show single data points while others do not. Please format consistently.

- Figure 5, this data is potentially interesting, but somewhat disconnected from the previous four figures. If I understand correctly, the conclusion is the same as Figure 4, in that the sequence identity of the GRE does not make any difference, correct? In that case, it could be moved to the supplement.

Additional comments:

- In the abstract it reads 'we used homology directed repair (HDR)-mediated genome editing to add a single GR binding site directly upstream of the transcriptional start site of four genes.' This is confusing. It would be helpful to mention Crispr engineering and to clearly state that one site was added to each of the four genes/loci tested.

- On p.3 the term 'peaks' in the third paragraph of the introduction presumably refers to binding sites identified by ChIP-Seq, please clarify.

- In the introduction or discussion, the discrepancy between large # of binding sites and small # of corresponding mRNA changes could also be due to: false positive ChIP signals, incorrect assignment of binding sites to target genes, single/few time points or conditions measured for the RNA, statistics and analysis parameters etc. Please discuss! Especially since the # of peaks varies greatly between experiments due to technical issues such as IP efficiency, this statement needs to be rephrased. (p. 3)

- p.4, the last paragraph of the introduction, is very general. It would be helpful to state the experimental approach, findings and conclusions more explicitly (Here we show... we find that...)

- A very minor point, Why did the authors use a non-perfect palindrome (AGAACAAttTGTACG)? Apparently it doesn't matter, but was it because this GRE previously showed the best activity?
- The manuscript would benefit from a graphical abstract.

The remaining open question still is what discriminates between a responsive, convertible gene and a non-recruiter.

Taken together, this potentially interesting study would be suitable for publication if the above comments can be addressed. The data appear to support the conclusions in most cases, but the above points should be discussed.

Reviewer #2 (Comments to the Authors (Required)):

In this manuscript, Thormann and colleagues study the impact of introduction and manipulation of GR consensus sequences in the human genome. The results show that a simple introduction of a single GR binding sites suffices to render that particular gene responsive to GR activation, albeit for a number of regions tested. Furthermore, they illustrate the changes in the GR motif, which clearly impacted activity in a plasmid-based reporter system, did not influence GR activity when inserted in the genome. This is a very well-written and interesting piece of work, which is of clear added value to the field. However, a number of issues would need some attention:

1. The H3K27ac and ATAC-seq data presented in the genome browser snapshots are represent the signal as observed in non-manipulated parental cells. Even though appealing, these data do not rule out the possibility that local epigenetic changes are found upon introduction of the sgRNA. While including ChIP-seq/ATAC-seq tracks for each condition may be a bit much to ask for, the authors should add QPCR experiments to illustrate whether there other epigenetic parameters are differentially impacted by introduction alone and/or GR activation.
2. please add statistics and/or individual datapoints for all barographs included in the paper.
3. For many of the genes studied, GR binding sites were observed rather close to the manipulated region. It is plausible that these sites don't impact the induction of GR-mediated effects upon gene editing, but the authors don't show this. A ChIP-QCPR for the proximal already existing GR sites (with or without gene editing) should illustrate whether GR binding at the adjacent regions is inert or also affected.

Reviewer #3 (Comments to the Authors (Required)):

Thormann et al demonstate that insertion of a single glucocorticoid receptor (GR) binding site just upstream from the transcription start site (TSS) of genes that are non-responsive to

glucocorticoids (GC) can convert some endogenous genes to GC-responsiveness, while others remain unresponsive. The ability to be activated correlated with creation of a new GR binding site and with the ability of a Cas9 activator protein directed at a site near the TSS to activate the genes. When variations of the GR binding site sequence were tested, there was little difference in their activities.

The study presented is commendable for the large number of clonal cell lines generated by the homologous repair CRISPR system. The conclusions drawn are well supported by the data. To my knowledge, this is the first study to investigate the use of gene editing to add a new regulatory feature to a gene. Although they were not able to determine exactly why two genes responded and two others did not, the experiments they performed ruled out certain possibilities and thus enhanced understanding of the criteria that determine which genes can be converted. The comparison of different GR binding sequences was also quite interesting; the lack of any difference in activity was surprising and informative given that previous reporter gene studies found a difference. Overall, this study advances knowledge of the importance of enhancer location and sequence in determining its activity. I have only a few relatively minor comments:

- 1) Statistical analyses are lacking for the quantitative RT-qPCR and ChIP data. Authors should provide relevant statistical analyses to validate any comparative statements they make in the manuscript.
- 2) Fig. 1 legend should specify that the data represent the status prior to gene editing.
- 3) Authors should explain that the abbreviation CGT (and the other GR binding sites they use for gene editing) refers to the gene that is spatially associated with the GR binding site that they are using for gene editing.
- 4) In the first Results paragraph, the sentence beginning "Second, because HDR efficiency ..." does not make logical sense. The first clause is about the relationship between HDR efficiency and distance from cut site, and this is given as a justification for placing the guide RNA binding site near the TSS.

Reviewer #1:

Comment 1. The authors find that two genes, *Il1b* and *Il1r2*, do respond and can be converted, while the other two, *Gypc* and *Vsig1*, do not. However, in Figure 1, it seems that there is already GR binding near these genes (but not at the promoter), and not at the non-responders. This could be the entire explanation for why the experiment works for those two. Please discuss (especially since this does not classify them as non-targets).

We agree with the reviewer that the presence of GR binding sites near the *IL1B* and *IL1R2* genes could explain why addition of a GBS is able to convert these genes into genes that change their expression upon GR activation (our working definition of GR targets). However, the presence of a nearby GR peak does not appear to be the entire explanation for gene-specific acquired gene regulation given that the *GYPC* gene, which cannot be converted, also harbors GR peaks near its TSS (Fig. 1B). Furthermore, the gene-specific activation by dCas9-SAM mirrors the pattern observed for GR even though these experiments were done in the absence of hormone and consequently absence of GR binding. To acknowledge the presence of nearby GR binding sites as possible explanation for the acquired GR-dependent regulation of the *IL1B* and *IL1R2* genes, we have added the following sentences to the discussion of the revised manuscript (page 8): "...The presence of endogenous GR binding sites near the *IL1B* and *IL1R2* genes (Fig. 1c, d) and the absence of a GR binding site near the *VSIG1* gene could also explain why the addition of a GBS is able to convert only *IL1B* and *IL1R2* into GR targets. However, the presence of a nearby GR binding site does not appear to be the entire explanation for gene-specific acquired gene regulation given that the *GYPC* gene, which cannot be converted, also harbors GR binding sites near its TSS (Fig. 1b)

Comment 2. Figure 2, A states n=1 for *Gypc*, but I assume it is n=3? E, f, g and h show the *Gilz* locus, but enrichment varies between 1.5 and 4 % input. Was this data normalized?

In contrast to the other genes for which we obtained at least 3 clonal lines, we only managed to isolate one single-cell-derived clonal line with an introduced GBS for the *GYPC* gene. So in this case, the data points shown are for biological replicates of this clonal line. This can also be deduced from supplementary Fig. 1A, where we only report the genotype for this one *GYPC* clonal line. To make this more obvious, we now explicitly mention that we only analyzed a single clonal line for the *GYPC* gene in the revised Fig. 2 legend.

Regarding the differences in the ChIP efficiencies for the pos. control locus (*GILZ*): When we started the project, we only edited the *IL1R2* gene and analyzed GR binding by ChIP. About a year later, we also decided to add a GBS at the TSS of other genes (*IL1B*, *GYPC* and *VSIG1*) and again analyzed GR binding by ChIP. For reasons we don't understand, the enrichment of our earlier ChIPs (about 4% input for the pos. control locus) was about twice that of the ChIPs we performed one year later for the *IL1B*, *GYPC* and *VSIG1* clonal lines (in the range of 1.5-2.5% of input for the pos. control locus). Importantly, comparisons between control and edited clonal lines were always derived from samples that were processed in parallel in the same experiment. Within experiments the control and edited clonal lines show comparable levels of enrichment at the pos. control locus, which is a prerequisite for a meaningful comparison of GR binding at the edited promoter. The ChIP data shown is the average % of input material that was immunoprecipitated in our ChIPs. This percentage was derived from comparing the ct values for each locus between input material and ChIP sample.

Comment 3. Figure 3, what are the controls for the dCas9 SAM? Is there equal recruitment to each locus?

As control for the specificity of activation, we compared RNA levels between gRNAs targeting the promoter and gRNAs that target the promoter of other genes. Specifically, we determined the RNA level of the targeted gene relative to *RPL19* (which served as an internal control) for cells transfected with the promoter-targeting gRNA. In addition, we determined the RNA level of the targeted gene for each of three guide RNAs that target the promoter of another gene. Finally, the fold regulation for the targeting gRNA was determined by dividing the normalized RNA level for the targeting gRNA by the average normalized RNA level for three non-targeting gRNAs. We have added a more detailed description of how the fold regulation was calculated to the materials and methods section of the revised manuscript.

We have not assayed if dCas9-SAM is recruited with equal efficiency to the promoter region of each of the genes assayed for Fig. 3. Therefore, differential recruitment might well explain the gene-specific activation by dCas9-SAM we observe. In support of this explanation, if we take the efficiency of GBS integration at the locus by HDR as a proxy for Cas9 recruitment, the observed activation somewhat mirrors the GBS addition efficiencies (13% and 35% efficiency respectively for *IL1R2* and *IL1B*, the genes with strong activation; 14% for *VSIG*, the gene with a marginal activation and only 6% for *GYPC*, the gene which was not activated by dCas9-SAM).

Comment 4: Figure 4c, how far is the wt peak from the promoter at the *Il1r2* locus? Can the ChIP-qPCR really discriminate between the two sites? It might be helpful to normalize the data to the control, to see fold enrichment. In general, some graphs show single data points while others do not. Please format consistently.

The distance between the wt peak and the *IL1R2* promoter region is approximately 1.8 kb. The fragment size of our ChIPed material is typically a couple of 100 bp, which should allow one to discriminate between these two loci. Accordingly, the ChIP seq data of Fig. 4c shows GR binding at the wt peak and no binding at the promoter for the unedited wt clonal control. We prefer not to display fold enrichment to avoid potential normalization issues, which can arise when the high ct values for the untreated control samples fluctuate between experiments. Throughout the manuscript, we consistently present average RNA expression levels for GBS variant groups as bar graphs with data points for individual clonal lines plotted as single data points. For the ChIP-qPCR data, we analyzed three biological replicates for a representative clonal line of each group and show the data as bar graph.

Comment 5. Figure 5, this data is potentially interesting, but somewhat disconnected from the previous four figures. If I understand correctly, the conclusion is the same as Figure 4, in that the sequence identity of the GRE does not make any difference, correct? In that case, it could be moved to the supplement.

The take home message of Fig. 5 is indeed that we did not observe obvious differences in activity between sequence variants of GR binding site. The difference with the results shown in Fig. 4 is that we analyzed an endogenous GR-bound locus rather than a synthetic sequence that we introduced at a gene normally not regulated by GR. In our opinion, the analysis of an endogenous GR-bound locus expands the scope of our findings and justifies keeping this figure as a main figure.

Comment 6. *In the abstract it reads 'we used homology directed repair (HDR)-mediated genome editing to add a single GR binding site directly upstream of the transcriptional start site of four genes. This is confusing. It would be helpful to mention Crispr engineering and to clearly state that one site was added to each of the four genes/loci tested.*

In the revised abstract, we now specify that the genome editing was CRISPR-based and we explicitly mention that one GR binding site was added for each of four genes tested. The relevant section now reads as follows: "...we used CRISPR/Cas-mediated homology directed repair to add a single GR binding site directly upstream of the transcriptional start site of each of four genes.

Comment 7. *On p.3 the term 'peaks' in the third paragraph of the introduction presumably refers to binding sites identified by ChIP-Seq, please clarify.*

We have addressed this ambiguity by revising various instances in the manuscript where we talk about "peak" to "ChIP-seq peak".

Comment 8. *In the introduction or discussion, the discrepancy between large # of binding sites and small # of corresponding mRNA chances could also be due to: false positive ChIP signals, incorrect assignment of binding sites to target genes, single/few time points or conditions measured for the RNA, statistics and analysis parameters etc. Please discuss! Especially since the # of peaks varies greatly between experiments due to technical issues such as IP efficiency, this statement needs to be rephrased. (p. 3)*

We agree with the reviewer that a more comprehensive discussion of possible explanations for the apparent disconnect between binding and regulation is important. Therefore, we have revised this paragraph of the introduction to the following:
".....GR can bind to tens of thousands of genomic binding sites, yet seems to regulate a smaller number of genes [4, 7, 12, 13]. Part of the discrepancy between GR binding and gene regulation might be technical, for example due to false positives in ChIP-seq peak calling and false negatives when the criteria for calling genes regulated are too stringent. Furthermore, gene regulation is typically only sampled at a few time points and relies on the analysis of steady state RNA which can yield false positives and false negatives e.g when changes in transcription rates are masked by changes in RNA stability. The discrepancy between GR binding and gene regulation might also be due to GR's inability to activate gene expression for a subset of occupied sites [14] and could reflect the inability of distal GR binding sites to contribute to gene regulation because they lack the physical proximity to the promoter of a gene.....".

Comment 9. *p.4, the last paragraph of the introduction, is very general. It would be helpful to state the experimental approach, findings and conclusions more explicitly (Here we show.. we find that.)*

In the revised manuscript, we have added the following sentence to complement the description of the experimental approach in the final paragraph of the introduction with a summary of our results:
"....Together, our studies reveal that addition of a single GBS can be sufficient to convert genes into GR targets without obvious differences in the level of activation between GBS variants.".

Comment 10. *A very minor point, Why did the authors use a non-perfect palindrome (AGAACAAttTGTACG)? Apparently it doesn't matter, but was it because this GRE previously showed the best activity?*

The rationale for picking this non-perfect palindromic sequence (CGT) is given in the second sentence of the results section: "...To increase our chances of observing GR-dependent regulation, we picked a GBS variant (CGT, a synthetic sequence matching the consensus motif), which showed the highest GR-dependent activation in previous studies [10, 11].....".

Comment 11. The manuscript would benefit from a graphical abstract.

In our opinion, Fig. 1A addresses this comment by providing a simple illustration of the experimental approach we used for our study.

Reviewer #2:

Comment 1. The H3K27ac and ATAC-seq data presented in the genome browser snapshots are represent the signal as observed in non-manipulated parental cells. Even though appealing, these data do not rule out the possibility that local epigenetic changes are found upon introduction of the sgRNA. While including ChIP-seq/ATAC-seq tracks for each condition may be a bit much to ask for, the authors should add QPCR experiments to illustrate whether there other epigenetic parameters are differentially impacted by introduction alone and/or GR activation.

We agree with the reviewer that we cannot rule out that the introduction of the GBS might influence H3K27ac levels, DNA accessibility or other chromatin characteristics at the edited locus. However, some of our findings argue against profound changes. First, introduction of the GBS does not influence basal expression levels of the edited genes (Fig. 2). Second, the activation by targeted recruitment of dCas9-SAM, which was measured in non-manipulated parental cells, follows the same gene-specific pattern of activation that was observed for GR. Finally, we have added an experiment where we compared H3K27ac levels at the *IL1R2* locus between wild type non-manipulated parental cells and a clonal line with an introduced GBS at the promoter of the *IL1R2* gene (Fig. S4B). The results from these ChIP-qPCR experiments indicate that H3K27ac levels at the edited *IL1R2* promoter locus do not change upon the addition of a GBS either in the presence or absence of GR activation by the addition of dex.

In the revised manuscript (page 8), we now acknowledge and discuss the possibility that the introduction of a GBS might change the local characteristics of the chromatin as follows: "...However, we cannot rule out that the GBS introduction might influence H3K27ac levels, DNA accessibility or other chromatin characteristics at the edited locus although some of our findings argue against profound changes. First, introduction of the GBS does not influence basal expression levels of the edited genes (Fig. 2). Second, the activation by targeted recruitment of dCas9-SAM, which was measured in non-manipulated parental cells, follows the same gene-specific pattern of activation that was observed for GR. Finally, ChIP experiments indicate that H3K27ac levels at the edited *IL1R2* promoter locus do not change upon addition of a GBS (Fig. S4b) either in the presence or absence of GR activation by the addition of dex.....".

Comment 2. Please add statistics and/or individual datapoints for all barographs included in the paper.

Following the suggestions by the reviewers, we have added statistical tests at various places to support comparative statements. Specifically, to support claims that *IL1R2* and *IL1B* acquire GR-dependent regulation upon GBS addition and that this is accompanied by acquired GR binding at the edited locus (Fig. 2). Furthermore, we added the results of statistical tests when we compare the activity of GBS variants (Fig. 4 and 5).

Comment 3. For many of the genes studied, GR binding sites were observed rather close to the manipulated region. It is plausible that these sites don't impact the induction of GR-mediated effects upon gene editing, but the authors don't show this. A ChIP-QCPR for the proximal already existing GR sites (with or without gene editing) should illustrate whether GR binding at the adjacent regions is inert or also affected.

In the revised manuscript, we have added a discussion of the potential role of existing GR binding sites near converted genes in facilitating the acquired GR-dependent regulation of the *IL1R2* and *IL1B* genes. Their possible role is also addressed in detail in response to comment 1 by reviewer #1. The effect of the added GBS on GR binding at an existing GR binding site (*IL1R2* GR wt peak) has been investigated for the *IL1R2* gene. The results are shown in Fig. 4c and indicate that binding at the existing *IL1R2* GR wt peak is not affected by the addition of a GBS at the *IL1R2* promoter.

Reviewer #3:

Comment 1: Statistical analyses are lacking for the quantitative RT-qPCR and ChIP data. Authors should provide relevant statistical analyses to validate any comparative statements they make in the manuscript.

Following the suggestions by the reviewers, we have added statistical tests at various places to support comparative statements. Specifically, to support claims that *IL1R2* and *IL1B* acquire GR-dependent regulation upon GBS addition and that this is accompanied by acquired GR binding at the edited locus (Fig. 2). Furthermore, we added the results of statistical tests when we compare the activity of GBS variants (Fig. 4 and 5).

Comment 2: Fig. 1 legend should specify that the data represent the status prior to gene editing.

In the revised manuscript, we have modified the description of the data presented in Fig. 1 to the following: “...**(b-e)** Tracks showing H3K27ac and GR ChIP-seq normalized tag density, ATAC-seq and RNA-seq reads for U2OS-GR cells (the non-edited parental cell line) that were treated as indicated.....”.

Comment 3: Authors should explain that the abbreviation CGT (and the other GR binding sites they use for gene editing) refers to the gene that is spatially associated with the GR binding site that they are using for gene editing.

In the revised manuscript, we have addressed this comment by adding a short statement regarding the source of each GBS when it is first mentioned (GILZ and FKBP5-2 are derived from GR-bound regions near those genes, CGT and PAL are synthetic sequences that match the consensus motif). See pages 4 and 6.

Comment 4: In the first Results paragraph, the sentence beginning "Second, because HDR efficiency ..." does not make logical sense. The first clause is about the relationship between HDR efficiency and distance from cut site, and this is given as a justification for placing the guide RNA binding site near the TSS.

To make it clear that the second criterion was chosen to increase the efficiency of HDR, we revised this sentence to the following (page 4): “.....Second, to increase our chances of obtaining correctly edited clones with a TSS-proximal GBS, we only considered genes with a possible guide RNA located ≤ 50 bp upstream of its TSS given that HDR efficiency decreases with increased distance between the cut site and the mutation [19].....”.

February 27, 2019

RE: Life Science Alliance Manuscript #LSA-2018-00283R

Dr. Sebastiaan H. Meijnsing
Max Planck Institute for Molecular Genetics, Berlin, Germany
Computational Molecular Biology
Innestrasse 63-73
Berlin 14195
Germany

Dear Dr. Meijnsing,

Thank you for submitting your revised manuscript entitled "Engineering genomic response elements to expand the glucocorticoid receptor target gene repertoire". Reviewer #1 evaluated the revision and appreciates the introduced changes. We would thus be happy to publish your paper in Life Science Alliance pending some minor final revisions:

- The reviewer proposes to add a graphical summary of your findings, and we would thus like to encourage you to do so.
- Please list 10 authors et al. in the reference list.
- Please add a callout in the manuscript text to figure 1A and 1F
- Please deposit the ATAC-seq and RNA-seq data and provide the accession numbers.

A. FINAL FILES:

-- Summary blurb (enter in submission system): A short text summarizing in a single sentence the study (max. 200 characters including spaces). This text is used in conjunction with the titles of

papers, hence should be informative and complementary to the title. It should describe the context and significance of the findings for a general readership; it should be written in the present tense and refer to the work in the third person. Author names should not be mentioned.

B. MANUSCRIPT ORGANIZATION AND FORMATTING:

Sincerely,

Andrea Leibfried, PhD
Executive Editor
Life Science Alliance
Meyerohofstr. 1
69117 Heidelberg, Germany
t +49 6221 8891 502
e a.leibfried@life-science-alliance.org
www.life-science-alliance.org

Reviewer #1 (Comments to the Authors (Required)):

This is the revised version of Thormann et al., Engineering genomic response elements to expand the glucocorticoid receptor target gene repertoire.

The authors test whether genes can be converted into GR targets by inserting a GRE sequence into the genome upstream (by Crispr/Cas).

The authors have addressed (most of) my previous concerns, and I'm satisfied with their response. Just one minor point: Figure 1a is a graphical abstract of the experimental design, but the point I raised was referring to a graphical representation of the outcome and the conclusions.

March 5, 2019

RE: Life Science Alliance Manuscript #LSA-2018-00283RR

Dr. Sebastiaan H. Meijsing
Max Planck Institute for Molecular Genetics, Berlin, Germany
Computational Molecular Biology
Innestrasse 63-73
Berlin 14195
Germany

Dear Dr. Meijsing,

Thank you for submitting your Research Article entitled "Engineering genomic response elements to expand the glucocorticoid receptor target gene repertoire". It is a pleasure to let you know that your manuscript is now accepted for publication in Life Science Alliance. Congratulations on this interesting work.

DISTRIBUTION OF MATERIALS:

Again, congratulations on a very nice paper. I hope you found the review process to be constructive and are pleased with how the manuscript was handled editorially. We look forward to future exciting submissions from your lab.

Sincerely,

Andrea Leibfried, PhD
Executive Editor
Life Science Alliance
Meyershofstr. 1
69117 Heidelberg, Germany
t +49 6221 8891 502
e a.leibfried@life-science-alliance.org
www.life-science-alliance.org